# Assessment of Physical Fitness and Risk Factors for the Occurrence of the Frailty Syndrome among Social Welfare Homes’ Residents over 60 Years of Age in Poland

**DOI:** 10.3390/ijerph19127449

**Published:** 2022-06-17

**Authors:** Antonina Kaczorowska, Katarzyna Szwamel, Małgorzata Fortuna, Agata Mroczek, Ewelina Lepsy, Aleksandra Katan

**Affiliations:** 1Institute of Health Sciences, University of Opole, 45-060 Opole, Poland; katarzyna.szwamel@uni.opole.pl (K.S.); agata.mroczek@uni.opole.pl (A.M.); ewelina.lepsy@uni.opole.pl (E.L.); 2Faculty of Medical Sciences and Technology, The Karkonosze University of Applied Sciences in Jelenia Góra, 58-503 Jelenia Góra, Poland; malgorzata.fortuna@kpswjg.pl; 3Rehabilitation Department, District Hospital Nachod, 547 69 Nachod, Czech Republic; akatan@wp.pl

**Keywords:** aging, physical functional performance, nursing homes, frailty syndrome, physical fitness, gait analysis

## Abstract

The study aimed at assessing physical fitness and occurrence of the frailty syndrome among social welfare homes’ residents as well as defining factors which determine the level of frailty and its occurrence. The examination included 198 residents (115 females and 83 males of average age 75.5 ± 10.21) and was carried out with the use of the Short Physical Performance Battery (SPPB) test with the following cut-off points: 0–6—frail, 7–9—pre-frail, 10–12—non-frail. The research additionally collected data regarding age, gender, number of chronic diseases, education level, type of prior work and current physical activity. In addition, the height and weight of the respondents were measured. The frailty syndrome was found in more than a half of the examinees (104; 52.53%), the pre-frailty state in 30.30% (*n* = 60) and 17.17% (*n* = 34) were non-frail. The average result of the SPPB test was 6.52 ± 2.73, which proves a moderate limitation of the sample group’s fitness. No significant differences were noted between female and male respondents (*p* = 0.27). The multifactorial linear regression model showed that independent and direct frailty syndrome predicators included age, number of chronic diseases and regular physical activity (*p* < 0.05). In conclusion, promoting and encouraging regular, age and interest-related forms of physical activity among seniors might foster the maintenance of their physiological reservoir and functional efficiency.

## 1. Introduction

Frailty is prevalent among elderly residents living in formal long- term care facilities [1]. One of the reasons of such a situation is that frailty and pre-frailty are significant predictors of nursing home placement among community-dwelling older adults [2]. Unfortunately, frailty syndrome is also an important predictor of mortality among older adults living in nursing homes [3]. Taking the reversible character of frailty into consideration, it is important to take a comprehensive view on frailty and carry out the appropriate interventions to prevent mortality and other adverse outcomes among social welfare homes’ residents [1,4].

In Poland in 2020, 67,000 of residents of stationary social welfare institutions were people over 60 years of age. Most of them (25,357 people, 24.1%) stayed in social welfare homes (SWH) [5]. However, an epidemiological study conducted in long-term care facilities in six European countries showed that Poland had one of the highest percentages of residents with poor functional and cognitive status [6].

Population ageing is one of the most significant demographic and social trends of the 21st century. In 2021, more than a fifth of the European Union (EU) population was 65 years of age or older, and this segment is projected to grow to 31.1% by 2100 [7]. Two of the most problematic expressions of population ageing are frailty and multimorbidity [8]. Therefore, promoting physical activity among older people in order to maintain their satisfactory health condition, physical activity and functional fitness and self-reliance has become one of the prioritized strategic areas established by WHO for European countries in 2016–2025 [9].

The prevalence of frailty ranges between 4% and 59% in elderly populations and is higher in women than in men [10]. More than 50% of the European population aged >50 years are pre-frail or frail (the overall prevalence of pre-frailty was 42.9% and frailty was 7.7%). The prevalence of frailty in Europe was estimated at approximately 3–15.6%, and in Poland it was—3.1% [11]. The studies showed that the prevalence of frailty and pre-frailty syndrome was of a higher incidence in inhabitants of formal long-term care (LTC) facilities than in people living in the community [12,13].

Frailty in aging marks a state of decreased reserves, resulting in increased vulnerability to adverse outcomes when exposed to stressors [14]. Functional reserve is essential to avoid stressors impacting function, and when intrinsic capacity and functional reserve are reduced the risk for additional disability is very high [15]. The most common concept of frailty is physical frailty. This concept includes the following criteria: unintentional weight loss (10 lbs in the past year), self-reported exhaustion, weakness (low grip strength), slow walking speed and low physical activity. Having at least three of them classifies as a frailty diagnosis [16]. The most common concept of frailty is physical frailty. This concept includes the following criteria: unintentional weight loss (10 lbs in past year), self-reported exhaustion, weakness (low grip strength), slow walking speed and low physical activity. Having at least three of them classifies one with a frailty diagnosis [16]. Chronic inflammation is likely to play a pivotal role in frailty, both directly and indirectly through other systems, such as the musculoskeletal, endocrine, and neurological systems [17]. Frailty increases health care expenditures and has a negative impact on older adults’ quality of life [1,18]. Frail elderly have been predisposed to functional deficits such as comorbidity and mortality because frailty reduces their ability to maintain overall homeostasis [19].

Nursing home residents are a particularly vulnerable to frailty [13,20]. The main determinants of being physically frail in nursing home residents‘ are: malnutrition [21], vitamin D deficiency [22,23], older age, female, living in a private institution, living with unknown person or living alone, having no regular exercise (≤2 times/week) and sedentary behaviour, poor self-reported health, lower socioeconomic status, lack of educational qualifications, obesity, being a smoker, and pain [24,25,26].

There is a lot of evidence which has proven that frailty among the elderly may be delayed or reversed. The following actions might prove effective: a multicomponent exercise programme, psychosocial intervention, cognitive stimulation, a combination of resistance exercise and protein supplementation [27,28,29].

The study aimed at (1) analysing and assessing physical fitness and the occurrence of the frailty syndrome among social welfare home residents as well as (2) defining factors significantly determining the level of the aspects mentioned above in the research group.

## 2. Materials and Methods

### 2.1. Study Design and Setting

This was a cross-sectional study from February 2019 to October 2019. The research was conducted in social welfare homes (SWH) in the Lower Silesia, Opolskie and Mazovian voivodships in Poland. The research was carried out in accordance to the Declaration of Helsinki and followed good clinical practice guidelines. The research project was approved by the Bioethics Committee of Opole Medical School (no KB/202/FI/2019). All participants gave written informed consent after explanation of the procedures involved. The STROBE guidelines (Strengthening the Reporting of Observational Studies in Epidemiology) were followed.

### 2.2. Participants

The study used a non-probabilistic sampling method. To calculate the minimal required number of participants for the sample, GUS data was used, which found 67,200 of those over 60 lived in a social welfare home SWH in Poland. With a confidence level of 95% and a margin of error of 5%, *p* = 50%, the minimum study sample was set at 382 subjects. Therefore, the information about the study and the request for the agreement were sent to 14 social welfare homes. The management of all the 14 facilities gave their consent for the examination. However, 165 out of 1320 residents had medical contraindications, 775 were unable to perform the fitness test, 158 did not comply with the age criterium and 24 did not agree to the examination (Figure 1).

Finally, 198 residents were qualified, including 115 women and 83 men aged 60–96. To be included in the study, participants were required to: (1) be of age of 60 years and over, (2) be able to move independently and take the fitness test, (3) not have any medical impediments, (4) be able to communicate verbally, and (5) provide voluntary written consent to participate in the study. The exclusion criteria comprised (1) acute injuries and infections, (2) recent myocardial infarction, (3) other medical impediments to research, (4) the lack of verbal contact, and (5) lack of written consent to participate in the study.

### 2.3. Measurement Tools

Initially, the respondents filled in the authors’ self-written questionnaire to collect data such as age, gender, number of chronic diseases, education, type of prior work and current physical activity. In addition, their height (to the nearest 0.5 cm) and weight (to the nearest 0.5 kg) were measured. Subjects were dressed in light clothing and stood barefoot, upright, and with eyes directed straight ahead when being measured. Body mass index (BMI) was calculated using participants’ height (in meters) and mass measurements (mass/height^2^). Using the World Health Organization (WHO) criteria (2000), BMI was used to categorize participants (underweight: <18.5 kg/m^2^; normal: 18.5–24.99 kg/m^2^; overweight: 25–29.9 kg/m^2^; obese: ≥30 kg/m^2^).

To assess physical condition the SPPB test was applied [30]. The test consists of three trials:

The assessment of strength and endurance of the lower extremities: 5-fold stand-ups off a chair with arms crossed on the chest. The time measured in seconds was recorded.

The assessment of static balance: a respondent is requested to keep their balance in three positions (side-by-side, semi-tandem stand and tandem balance stand). Each next position is performed if no complications with the previous one occur and an examinee is able to withstand it for 10 s.

The assessment of walking speed: walking 4 metres at a normal pace. If examinees walk with some orthopaedic aids, they use it during the examination. The time of the trial was recorded.

The results recorded for each participant were compared with normative data and assigned to 0–4 points for each trial. The overall test score ranged from 0–12 points [30,31,32]. The SPPB test scoring was used to assess the occurrence of the frailty syndrome. Regarding the threshold score for frailty, older adults who score ≤ 9 on the SPPB are most likely to be classified as frail [33], and are at risk of losing the ability to walk 400 m [34] (predictive validity). An SPPB score of ≤9 has the most desirable sensitivity (92%), specificity (80%) and greatest area under the curve (AUC = 0.81) for identifying frail adults [35]. In order to classify participants as frail, pre-frail and non-frail, the following cut-offs were used: SPPB 0–6 (frail), SPPB 7–9 (pre-frail), SPPB 10–12 (non-frail) [36]. The measurements and the SPPB test were supervised and performed by the same professionals. They were performed in the morning hours, in SWH common rooms and with the use of a standard chair to assure the same research conditions.

### 2.4. Statistical Methods

The analysis of the quantitative variables was made by calculating the mean (M), standard deviation (SD), median (Me) and quartiles (Q1, Q3). The qualitative variables’ data was assessed by calculating numbers and percentage of occurrence for each value individually. The comparison of qualitative variables’ values in groups were counted with the use of a chi-square test (with Yate’s correction for 2 × 2 tables) or a Fisher’s test. The comparison of quantitative variables in two groups was carried out with a Mann-Whitney test, while the comparison of quantitative variables’ values in three or more groups was counted with Kruskal-Wallis’ test. After statistically significant differences had been observed, the post-hoc analysis was applied with the use of Dunn’s test to identify statistically and significantly different groups. The comparison between qualitative variables’ values in three or more subsequent measurements was calculated with Friedman’s test. After statistically significant differences had been revealed, the post-hoc analysis (Wilcoxon’s test for related pairs with Bonferroni correction) was applied so as to identify statistically different measurements. The correlations between quantitative variables were made with the use of Spearman’s correlation coefficient. The multifactorial analysis of the impact of many variables on one qualitative variable was assessed with the linear regression method. The results were presented in the form of regression model parameter values with a 95% confidence interval. The significance level for the analysis was 0.05. The analysis was calculated in programme R, version 4.1.2 [37].

## 3. Results

### 3.1. Descriptive Data

Compared to men, women were statistically significantly older (*p* < 0.001), had more numbers of chronic diseases (*p* < 0.001), and had higher BMI values (*p* = 0.028). Statistically significant differences were also observed between the genders in terms of education (*p* = 0.022) and prior work (*p* = 0.007) (Table 1).

### 3.2. Main Results

The mean of the SPPB test score was 6.52 ± 2.73 and the median was 6, which proves a moderate limitation of the group’s fitness. The best statistically significant result was obtained in the walking speed at the 4 m distance trial (2.44 ± 1.07; *p* < 0.001) compared to other trials (Table 2).

The prevalence of frailty among the residents of SWH was found in over half of the respondents (*n* = 104; 52.53%), both pre-frail (*n* = 60; 30.30%) and non-frail (*n* = 34; 17.17%). There were no statistically significant differences between groups of women and men (Table 3).

Gender differentiates the results of the SPPB test, however only in a ‘walking speed’ trial, where the statistically significantly higher score was in men rather than in women (2.65 ± 1.12 vs. 2.29 ± 1.01; *p* = 0.022). The level of activity declared by the respondents also significantly differentiated the results of the SPPB test. The overall SPPB score and trial scores were significantly higher in the physically active group than in all other groups (*p* < 0.001). However, there were no statistically significant differences when taking into account such variables as education or prior work (Table 4).

A weak (r = −0.2), but statistically significant negative correlation between age and the overall number of the SPPB test points (*p* = 0.003) and the points at the ‘standing off a chair’ (*p* < 0.001) and ‘walking speed at 4 m distance’ trials (*p* = 0.003) were found. It might be concluded that the higher the age, the lower the score at the trials mentioned above. We also found a moderate (r = −0.5) statistically significant negative correlation between number of chronic diseases and SPPB test (*p* < 0.001) and ‘walking speed at a 4 m distance’ (*p* < 0.001). Another weak statistically significant negative correlation was found between number of chronic diseases and ‘standing off a chair’ (r = 0.3; *p* < 0.001) and balance test (r = 0.4; *p* < 0.001). The more chronic diseases the residents of SWH suffered from, the lower the scores in all examined aspects (Table 5).

The multifactorial model of linear regression showed that significant (*p* < 0.05), independent and direct predictors of the frailty syndrome included age, number of chronic diseases and regular physical activity. Each next year of age decreased the SPPB test score by 0.042 pts. on average (regression parameter −0.042). Each additional chronic disease decreased the SPPB test score by 1.071 pts. on average (regression parameter −1.071), and physical activity increased the score by 2.3 pts. compared to the complete lack of activity (regression parameter 2.3) (Table 6).

Age, the number of chronic diseases and regular physical activity proved to be significant (*p* < 0.05), as did direct predictors of the ‘standing off a chair’ and ‘walking speed at 4 m distance’ trials, while prior combination work, number of chronic diseases and regular physical activity mattered at the ‘balance test’ trial. In the ‘standing off a chair’ trial, each next year of age decreased the trial score by 0.028 pts. on average (regression parameter −0.028), and each additional chronic disease decreased the score by 0.272 pts. on average (regression parameter −0.272) and physical activity increased the score by 0.723 pts. on average compared to the complete lack of activity (regression parameter 0.723). In the ‘balance test’ trial, prior combination work enhanced the trial score by 0.56 pts. while compared to physical work (regression parameter 0.56), each additional chronic disease reduced the score by 0.421 pts. on average (regression parameter −0.421), and physical activity increased the score by 0.668 pts. compared to complete lack of activity (regression parameter 0.668). In the ‘walking speed at 4 m distance’ trial, each next year of age reduced the trial score by 0.015 pts. on average (regression parameter 0.015), and each additional chronic disease reduced the score by 0.38 pts. (regression parameter −0.38), and physical activity increased the score by 0.904 pts. compared to the complete lack of activity (regression parameter 0.904) (Table 7).

## 4. Discussion

### 4.1. Key Results and Interpretation

The primary aim of the research was to analyse and assess physical activity and the occurrence of the frailty syndrome among the SWH residents. The average score of the sample group proves their moderate limitation with regard to fitness. The best scores were noted at the ‘walking speed at 4 m distance’ trial, whereas the lowest were found at the ‘5-fold-standing off a chair’, which measured the strength and endurance of lower extremities, or the ‘balance test’. The overall SPPB test results, ‘standing off a chair’ and ‘balance test’ trials’ scores in female and male groups did not differ from one another to any significant degree. Male respondents noted significantly better results at the ‘walking speed at 4 m distance’ trial. Similar results were noted by Guede Rojas et al. They used a senior fitness test and found that the elderly male group achieved better results than the female one in a ‘2-min marching test’ [38]. De Amorim at al. indicated that frailty prevalence is significantly higher among women than men [39].

Such relatively low results achieved by the respondents in the SPPB test might be related to the place of residence, namely SWH. Most physical activity performed by elderly people is connected with their household duties and daily routine. SWH residents lack this kind of daily activity. The performing of daily chores in SWH is highly limited. It is possible that reducing their daily regular activities results in limited physical fitness and significantly lower scores at the ‘walking speed’ trial, especially among women.

The limitations of opportunities in performing daily duties among the elderly, resulting from low levels of fitness or physical endurance, are closely related to the lack of regular physical activity. According to Fisher et al., a low level of physical activity was connected with the fact of dwelling in a SWH and the level of activity decreased with the age of seniors [40]. Residing in nursing institutions means leading a sedentary lifestyle [41]. Barber et al. assessed SWH residents’ daily activity for seven days with the use of an accelerometer. The results showed that the level of activity among the examinees was very low: they spent 79% of the day in a sitting position [42].

Low levels of physical activity in Polish seniors may be related to cultural and social factors as well. The current generation of seniors acquired their habits and behavioural patterns, as well as the ones connected with physical activity, in communist Poland. Polish and other post-communist countries’ seniors lived in a different cultural context than their West European or North American counterparts, where the idea of active aging was thoroughly grounded. In Poland, senility was traditionally considered as the time of well-deserved rest. Also a small fraction of Bohemian seniors take part in sports activities or other forms of physical exercise [43]. The results are collected in the research project entitled ‘Bridging the East–West Health Gap’, which aimed at examining health condition, attitude and pro-health behaviours in adults from selected Central-Eastern and Western countries, and indicated a huge diversity of physical activity levels in each country. The greatest proportion of physically active respondents was found in West European countries (30.2% Finland, 23.7% Spain) whereas the smallest was in post-communist ones (6.4% Poland, 12.3% Hungary) [44].

Attention ought to be paid to the high proportion of frail and pre-frail respondents found in the self-reported study and the fact that only 17.17% were non-frail. It might be concluded that most of the elderly residents of SWH are at risk of the frailty syndrome. The research by Furtado et al. confirms that institutionalized women, who are found less physically active and not self-reliant, are particularly prone to frailty syndrome occurrence [45]. The issue of frailty in SWH residents was also studied by Kaczorowska et al. The researchers examined 85+ women residing SWH. There was no non-frail individual found among 17 women [20]. The frailty syndrome reduces an elderly person’s self-reliance. It leads to an increased vulnerability to unfavourable health-related incidents such as falls, hospitalizations, disability, institutional residence or death [16,46,47]. The correlation between the frailty syndrome and falls in the elderly, assessed on the basis of low SPPB test results, was presented by various authors [47]. Early recognition of the risk of frailty is incredibly crucial, as thanks to the multidirectional prophylaxis there is a possibility to prevent it and improve a patient’s condition [48].

More and more research currently makes use of the SPPB test to assess frailty syndrome occurrence. Pritchard et al. examined patients from a geriatric out-patient clinic at the Centre for Healthy Aging in Canada according to Fried’s phenotype method with the use of the SPPB test. They achieved similar results to the self-reported ones. After the application of the SPPB test, they found out that 50% of the patients were frail, 35% were pre-frail and 15% were non-frail. Taking into account the Fried’s phenotype method, 35% were frail, 57% pre-frail and 7% were non-frail. There was fair to moderate agreement between methods for determining which participants were frail and pre-frail [48]. Danilowich et al. researched seniors in a care home in Illinois with the same test and free online calculator, SHARE-FI. Their online results also resembled the self-reported ones: 45% of the respondents were found to be frail, 35% were pre-frail and 20% were non-frail. The results of the SPPB test revealed that 69% were frail, 28% were pre-frail and 3% were non-frail patients. There was fair to moderate, but statistically significant agreement between these measures [49]. A Spanish study was conducted among over 65year-olds and those respondents living independently. The frailty syndrome was diagnosed with the use of the Frailty Trail Scale (FTS) and physical fitness was assessed with the SPPB test. The authors found a significantly adverse correlation between the results of both measures. A lower score on the SPPB test was related to a higher score on the FTS test and higher intensity of frailty syndrome [50].

The following aim of the study was to determine the factors importantly influencing the level of physical fitness and the occurrence of the frailty syndrome in the research group. It was revealed that demographic variables affected the SPPB test results. Gender was statistically determinant only in the ‘walking speed at 4 m distance’ trial, although men achieved slightly better results in all trials as well. Other demographic variables, such as education or prior work experience, did not affect physical fitness to any significant degree. Some authors, however, report that frailty prevalence is significantly higher among those having a low educational level and those whose job was predominantly physical. These findings may suggest that work factors could explain the incidence of frailty syndrome [39]. This was not confirmed by the results of our analyses. This would require further research. On the other hand, the factor that significantly differentiated the results was physical activity. The respondents who declared being physically active achieved higher overall scores as well as in each individual trial than those who rarely or never undertook physical activity. Taking into consideration correlations between age, number of chronic diseases and BMI, the first two variables correlated adversely with the level of fitness. The multifactorial model of linear regression also showed that age, number of chronic diseases and regular physical activity were independent and direct predictors of frailty. The research by other authors found some other variables as strong predictors, namely advanced age elderly, osteoarticular disease, as well as history of hospitalization and falls in the last twelve months [51]. The research by Miller et al. showed that a lower score at the SPPB test was linked to higher age, falls and chronic diseases such as diabetes, sight disorders and kidney issues [52].

The decline of physical activity and functional fitness related with age among elderly men and women was confirmed in the study by Milanovic et al. [53] in which young elderly (60–69 years of age) achieved better results than old elderly (70–80) in almost all the trials of physical fitness assessed with the use of the Senior Fitness Test. Moreover, the study concluded that the decrease of physical activity and functional fitness is caused by a natural aging process. The research of Delbari at al. shows that only the age predictor variable has a statistically significant effect on the occurrence of frailty and, indeed, the frequency of frail older adults significantly increases with age. This result was supported by other studies [29,54,55]. We know that frailty risk increases in association with age, which could be due to the biological rather than the chronological age of individuals. There is consequently an erosion of the homeostatic reserve and vulnerability to disproportionate changes in health status after relatively minor stress events. There is a continuous loss of strength and aerobic resistance, which causes a decrease in functional independence and makes the older adult frail. In general, frailty is superior to age in identifying at-risk older people [56,57].

A vast number of researchers highlight the importance of physical activity and its direct influence on the level of fitness in the elderly as well as the occurrence of the frailty syndrome. The differences in scores of physical fitness between sedentary and active lifestyle groups were confirmed by the study of Silva et al. [58]. The data related to the correlation between sedentary lifestyle or the level of physical activity and physical fitness among elderly patients revealed that the active group achieved higher scores on the Senior Fitness Test than the inactive group. The authors concluded that future prospective research ought to assess the level of physical activity more objectively and discover the causal links between the level of physical activity and fitness in the elderly. To maximize the benefits of physical activity, older people should be encouraged to break down their daily sedentary routine and avoid long-lasting sedentary periods. According to the research by Silva et al., the frailty syndrome is more common among older people who are insufficiently active and spend most of their time sitting, even when considering socio-demographic factors [59].

The scientific data acknowledges that the level of physical activity usually decreases with age and is connected with the decline of functional fitness [52]. It confirms the assumption that the level of physical activity is affected by the aging process and leads to the reduction of functional fitness. The level of physical activity influences the maintenance and the increase of physical fitness [60], and every form of physical activity is better than none [61,62]. The research [63] also noted that lifestyle behaviours such as physical activity may help manage the level of frailty. Adversely, a sedentary lifestyle is connected with frailty regardless of physical activity. Prolonged sitting comes with higher risks of mortality in frail elderly people. Conversely, the research by Billot et al. into the mobility behaviours in the frail elderly suffering from sarcopenia revealed that one of the most common features of aging is the decline of functional skills. Physical frailty and sarcopenia are characterised by weakness, slowness and reduced muscle mass with maintained independent walking skills. One of the strategies which showed some benefits in fighting the mobility loss and its consequences in the elderly is physical activity [64]. Sarcopenia and frailty have mutual aetiology, but aging is connected with a changed signalization of redox in the skeletal muscles. Modifiable risk factors improve protein synthesis and prevent muscle loss with age. Therefore, physical activity plays a crucial role in preventing these processes [65].

The profession and previous type of work affect the accumulated cognitive reserves. In everyday life, we use only a small part of our cognitive resources. With age, the intellectual abilities are impaired, and this reserve is activated and allows for the compensation of the emerging deficits, which guarantees the maintenance of good cognitive functioning until old age [66]. Our own research showed that the type of prior work and physical activity were important independent direct predictors of the result of the static equilibrium test. A prior combination of physical and mental work and systematic physical activity has a positive effect on balance. Research by Yokoyama et al. showed that two-task cognitive-motor training was more beneficial than just motor training in improving cognitive functions in sedentary elderly people [67]. Similarly, in the studies by Hagovska and Olkeszyova, significant relationships between balance, walking speed and cognitive functions were recorded among the elderly with cognitive impairment who participated in cognitive-motor training [68].

To conclude, aging results in the body fat, reduction of muscle strength, and lower levels of flexibility, agility, and endurance. However, the process of aging is natural and inevitable, and an appropriate level of physical activity might slow down the loss of functional and physical skills and help maintain healthy lifestyles in the elderly [69].

### 4.2. Strenghts of the Study

The standardized testing tool, the SPPB test, was used, which is highly sensitive at identifying frail individuals and correlates well with other methods which assess the phenomenon, such as Fried’s phenotype method, the free online calculator SHARE-FI and the Frailty Trait Scale.

### 4.3. Limitations

This research has some limitations. As we mentioned in the methodology section, our study lasted from February 2019 to October 2019. In this period we analyzed 198 participants. We are aware that ultimately we should examine 382 participants. Following previous research by Hamilton et al. [70] and Mizumoto et al. [71] we decided to intentionally suspend our research during the winter period. These authors have reported that the activity range for adults decreases during the winter season compared with that reported during the summer season [70,71]. If we were to continue the research in the winter of 2019/2020, we could have obtained much worse results in the SPPB test. We wanted to continue our research in March 2020, but the first cases of patients with COVID-19 in Poland were reported at that time. From 4 March to 30 April 2020, a total of 12,877 laboratory-confirmed COVID-19 cases were registered in Poland. The highest rates for COVID-19 were recorded in the Lower Silesia Province, Silesia and Mazovia [72]. This was the area of our research. As Raciborski et al. reported, the main setting of SARS-CoV-2 transmission was nursing homes (outbreaks of infection in long-term care facilities) [72]. Therefore, we have not obtained consent to continue our research on the forms of the management of social welfare homes. Therefore, we finally decided to analyze the data from 198 participants.

The second limitation of the study was the fact that the nutrition style in SWH was not taken into account, and no standardized measure was applied to assess the examinees’ physical activity. A further limitation may stem from the fact that the correlation between place of residence of seniors and the level of their fitness may be adverse. There is a strong need for further research to determine the factors affecting frailty syndrome occurrence in institutionalized people.

## 5. Conclusions

Functional fitness of elderly SWH residents was limited to a moderate degree. Limiting physical fitness causes a decrease in independence in everyday activities. Systematic physical activation of the inhabitants of nursing homes would limit this unfavorable phenomenon. A large proportion of the SWH seniors were frail or pre-frail, which proves their susceptibility to disability, lower immunity to stressors and decreased adaptational or physiological body reservoirs. The consequence of the frailty syndrome is disability and frequent hospitalizations. Therefore, prophylaxis and activities aimed at improving the condition of people diagnosed as frail should be introduced in social welfare homes.

Direct risk factors of the frailty syndrome were age, number of chronic diseases and low level of physical activity. Proper treatment of chronic diseases as well promoting regular, age and interest-related forms of physical activity among seniors as well as encouraging SWH residents to take part in physical activity classes may facilitate physiological and functional fitness reservoirs’ maintenance.

## Figures and Tables

**Figure 1 ijerph-19-07449-f001:**
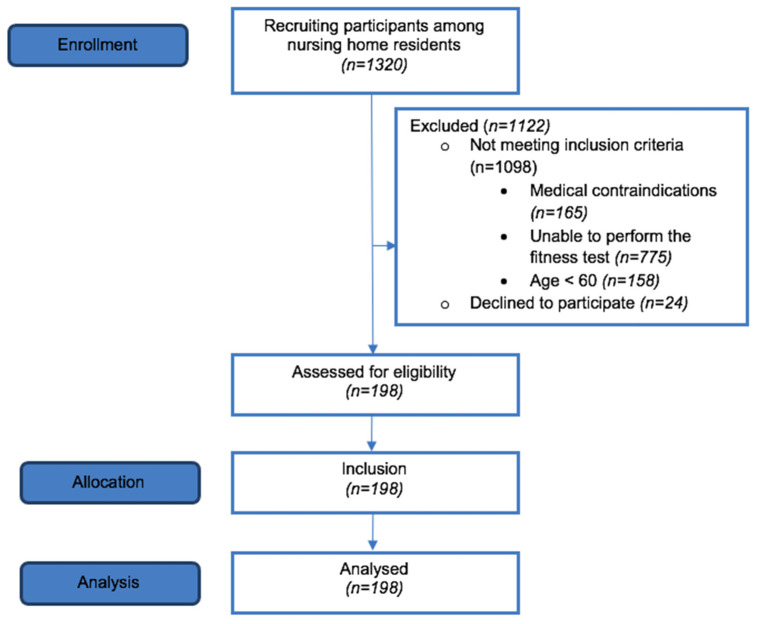
Flow diagram of the study.

**Table 1 ijerph-19-07449-t001:** The characteristics of a sample group.

Parametr		Gender			*p*
		Women (N = 115)	Men (N = 83)	Total (N = 198)	
Age [years]	M ± SD	77.83 ± 10.17	72.27 ± 9.39	75.5 ± 10.21	*p* < 0.001 *
	Me	80	69	78	
	Q1–Q3	70.5–86	64.5–80.5	66–84	
Number of chronic diseases	M ± SD	2.75 ± 1.09	2.3 ± 0.97	2.56 ± 1.06	*p* = 0.001 *
	Me	3	2	2	
	Q1–Q3	2–3	2–3	2–3	
Education	No	4 (3.48%)	2 (2.41%)	6 (3.03%)	*p* = 0.022 *
	Primary	53 (46.09%)	28 (33.73%)	81 (40.91%)	
	Vocational	13 (11.30%)	25 (30.12%)	38 (19.19%)	
	Secondary	38 (33.04%)	24 (28.92%)	62 (31.31%)	
	High	7 (6.09%)	4 (4.82%)	11 (5.56%)	
Prior work	Physical	68 (59.13%)	67 (80.72%)	135 (68.18%)	*p* = 0.007 *
	Combination of physical and mental	15 (13.04%)	7 (8.43%)	22 (11.11%)	
	Mental	22 (19.13%)	8 (9.64%)	30 (15.15%)	
	No	10 (8.70%)	1 (1.20%)	11 (5.56%)	
Physical activity	No	41 (35.65%)	32 (38.55%)	73 (36.87%)	*p* = 0.275
	Rare	35 (30.43%)	17 (20.48%)	52 (26.26%)	
	Yes	39 (33.91%)	34 (40.96%)	73 (36.87%)	
BMI [kg/m^2^]	M ± SD	27.69 ± 5.63	26.01 ± 4.84	26.99 ± 5.37	*p* = 0.028 *
	Me	27.27	25.26	26.3	
	Q1–Q3	23.86–31.56	22.58–28.52	23.2–30.1	
BMI interpretation	Underweight	4 (3.48%)	2 (2.41%)	6 (3.03%)	*p* = 0.156
	Standard	37 (32.17%)	37 (44.58%)	74 (37.37%)	
	Overweight	37 (32.17%)	28 (33.73%)	65 (32.83%)	
	Obesity	37 (32.17%)	16 (19.28%)	53 (26.77%)	

Legend: *p*-Mann-Whitney’s test for quantitative variables, Chi-squared or exact Fisher’s test for qualitative variables, * statistically significant difference (*p* < 0.05).

**Table 2 ijerph-19-07449-t002:** The statistical characteristics of the SPPB test results.

Trial	N	M	SD	Me	Min	Max	Q1	Q3	*p*
SPPB test	198	6.52	2.73	6	1	12	4.25	9	
Standing off a chair (A)	198	1.95	1.2	2	0	4	1	3	*p* < 0.001 *
Balance test (B)	198	2.13	1.07	2	0	4	1	3	
Walking speed at a 4 m distance (C)	198	2.44	1.07	2	1	4	2	3	C > A.B

Legend: *p*-Friedman’s test + post-hoc analysis (Wilcoxon’s test for related pairs with Bonferroni correction). * statistically significant difference (*p* < 0.05).

**Table 3 ijerph-19-07449-t003:** The proportion of frail, pre-frail and non-frail residents in a sample group.

Frailty Syndrome Ranges	Gender			*p*
	Women (N = 115)	Men (N = 83)	Total (N = 198)	
Frail	66 (57.39%)	38 (45.78%)	104 (52.53%)	*p* = 0.27
Pre-frail	31 (26.96%)	29 (34.94%)	60 (30.30%)	
Non-frail	18 (15.65%)	16 (19.28%)	34 (17.17%)	

Legend: *p*-chi-squared test.

**Table 4 ijerph-19-07449-t004:** Statistical characteristics of the SPPB test results in correlation to gender, education, prior work and level of physical activity.

**Test**		**Gender**				** *p* **
		**Women (N = 115)**		**Men (N = 83)**		
SPPB test	M ± SD	**6.28 ± 2.68**		**6.86 ± 2.78**		*p* = 0.128
	Me	6		7		
	Q1–Q3	4–8		5–9		
Standing off a chair	M ± SD	1.87 ± 1.2		2.06 ± 1.2		*p* = 0.220
	Me	1		2		
	Q1–Q3	1–3		1–3		
Balance test	M ± SD	2.11 ± 1.02		2.16 ± 1.13		*p* = 0.797
	Me	2		2		
	Q1–Q3	1–3		1–3		
Walking speed at a 4 m distance	M ± SD	2.29 ± 1.01		2.65 ± 1.12		*p* = 0.022 *
	Me	2		3		
	Q1–Q3	1–3		2–4		
**Test**		**Education**				** *p* **
		**No, primary (N = 87)**	**Vocational** **(N = 38)**	**Secondary (N = 62)**	**High** **(N = 11)**	
SPPB test	M ± SD	6.41 ± 2.49	6.92 ± 2.84	6.71 ± 2.79	4.91 ± 3.56	*p* = 0.304
	Me	6	7	6	5	
	Q1–Q3	5–8	5–9	5–9	2–7.5	
Standing off a chair	M ± SD	1.93 ± 1.14	2.05 ± 1.18	2.03 ± 1.28	1.27 ± 1.27	*p* = 0.212
	Me	2	2	2	1	
	Q1–Q3	1–3	1–3	1–3	0.5–1.5	
Balance test	M ± SD	2.1 ± 1.01	2.26 ± 1.08	2.19 ± 1.04	1.55 ± 1.51	*p* = 0.356
	Me	2	2	2	2	
	Q1–Q3	1–3	2–3	1–3	0–2	
Walking speed at a 4 m distance	M ± SD	2.36 ± 1.03	2.66 ± 1.07	2.48 ± 1.1	2.09 ± 1.14	*p* = 0.352
	Me	2	3	2.5	2	
	Q1–Q3	2–3	2–4	2–3	1–3	
**Test**		**Prior Work**				** *p* **
		**Physical** **(N = 135)**	**Combination of Physical and Mental** **(N = 22)**	**Mental** **(N = 30)**	**No** **(N = 11)**	
SPPB test	M ± SD	6.53 ± 2.5	6.55 ± 2.65	6.47 ± 3.69	6.55 ± 3.01	*p* = 0.997
	Me	6	6	6	6	
	Q1–Q3	5–8	5–9	3–10	4.5–9	
Standing off a chair	M ± SD	1.98 ± 1.15	1.86 ± 1.17	1.87 ± 1.48	2 ± 1.26	*p* = 0.811
	Me	2	1	1	2	
	Q1–Q3	1–3	1–2	1–3	1–2.5	
Balance test	M ± SD	2.1 ± 1.01	2.23 ± 1.11	2.17 ± 1.26	2.27 ± 1.27	*p* = 0.934
	Me	2	2	2	2	
	Q1–Q3	1–3	2–3	1–3	1–3.5	
Walking speed at a 4 m distance	M ± SD	2.46 ± 1.02	2.45 ± 1.1	2.4 ± 1.3	2.27 ± 1.01	*p* = 0.949
	Me	2	2	2.5	2	
	Q1–Q3	2–3	2–3	1–4	1.5–3	
**Test**		**Physical Activity**				** *p* **
		**No-A** **(N = 73)**	**Rare-B** **(N = 52)**		**Yes-C** **(N = 73)**	
SPPB test	M ± SD	5.53 ± 2.38	5.98 ± 2.62		7.89 ± 2.61	*p* < 0.001 * C > B.A
	Me	5	5.5		8	
	Q1–Q3	4–7	4–8		6–10	
Standing off a chair	M ± SD	1.68 ± 1.12	1.67 ± 1.13		2.41 ± 1.21	*p* < 0.001 * C > A.B
	Me	1	1		2	
	Q1–Q3	1–2	1–3		1–4	
Balance test	M ± SD	1.82 ± 0.98	2.02 ± 0.98		2.52 ± 1.11	*p* < 0.001 * C > B.A
	Me	2	2		2	
	Q1–Q3	1–2	1–2		2–3	
Walking speed at a 4 m distance	M ± SD	2.04 ± 0.92	2.25 ± 1.05		2.97 ± 1.01	*p* < 0.001 * C > B.A
	Me	2	2		3	
	Q1–Q3	1–3	1–3		2–4	

Legend: *p*-Mann-Whitney’s test (gender), *p*-Kruskal-Wallis test (education, prior work), *p*-Kruskal-Wallis test + post hoc analysis (Dunn’s test) (physical activity), * statistically significant difference (*p* < 0.05).

**Table 5 ijerph-19-07449-t005:** Correlations between the SPPB test results and age, number of chronic diseases and BMI.

Variable	Test	Spearman’s Corellation Coefficient
Age [years]	SPPB test	r = −0.213. *p* = 0.003 *
	Standing off a chair	r = −0.265. *p* < 0.001 *
	Balance test	r = −0.041. *p* = 0.565
	Walking speed at a 4 m distance	r = −0.211. *p* = 0.003 *
Number of chronic diseases	SPPB test	r = −0.489. *p* < 0.001 *
	Standing off a chair	r = −0.351. *p* < 0.001 *
	Balance test	r = −0.434. *p* < 0.001 *
	Walking speed at a 4 m distance	r = −0.466. *p* < 0.001 *
BMI [kg/m^2^]	SPPB test	r = −0.074. *p* = 0.3
	Standing off a chair	r = −0.041. *p* = 0.569
	Balance test	r = −0.039. *p* = 0.585
	Walking speed at a 4 m distance	r = −0.132. *p* = 0.063

Legend: * statistically significant difference (*p* < 0.05).

**Table 6 ijerph-19-07449-t006:** Direct predictors of the frailty syndrome–the multifactorial analysis.

SPPB Test					
Feature		Parameter	95%CI		*p*
Gender	Women	ref.			
	Men	−0.104	−0.803	0.596	0.771
Age	[years]	−0.042	−0.075	−0.009	0.013 *
Education	No, primary	ref.			
	Vocational	0.057	−0.851	0.965	0.902
	Secondary	−0.783	−1.803	0.237	0.134
	High	−1.037	−2.893	0.82	0.275
Prior work	Physical	ref.			
	Combination of physical and mental	1.045	−0.252	2.342	0.116
	Mental	0.924	−0.331	2.178	0.151
	No	0.101	−1.346	1.548	0.891
Number of chronic diseases		−1.071	−1.395	−0.747	<0.001 *
BMI	[kg/m^2^]	0.022	−0.039	0.084	0.477
Physical activity	No	ref.			
	Rare	0.704	−0.107	1.514	0.091
	Yes	2.3	1.553	3.047	<0.001 *

Legend: *p*-multifactorial linear regression, * statistically significant difference (*p* < 0.05).

**Table 7 ijerph-19-07449-t007:** Direct predictors of ‘standing off a chair’, walking speed at 4 m distance’ and ‘balance test’ trials.

**Standing Off a Chair**					
**Feature**		**Parameter**	**95%CI**		** *p* **
Gender	Women	ref.			
	Men	0.083	−0.197	0.362	0.563
Age	[years]	−0.015	−0.028	−0.002	0.03 *
Education	No, primary	ref.			
	Vocational	0.078	−0.285	0.441	0.674
	Secondary	−0.241	−0.649	0.167	0.249
	High	−0.035	−0.777	0.707	0.926
Prior work	Physical	ref.			
	Combination of physical and mental	0.339	−0.179	0.858	0.201
	Mental	0.202	−0.3	0.704	0.431
	No	−0.035	−0.613	0.544	0.907
Number of chronic diseases		−0.38	−0.51	−0.251	<0.001 *
BMI	[kg/m^2^]	−0.002	−0.027	0.023	0.874
Physical activity	No	ref.			
	Rare	0.313	−0.011	0.637	0.06
	Yes	0.904	0.605	1.202	<0.001 *
**Balance Test**					
**Feature**		**Parameter**	**95%CI**		** *p* **
Gender	Women	ref.			
	Men	−0.107	−0.402	0.188	0.479
Age	[years]	0.001	−0.013	0.015	0.889
Education	No, primary	ref.			
	Vocational	0.157	−0.226	0.54	0.422
	Secondary	−0.336	−0.766	0.094	0.128
	High	−0.506	−1.289	0.277	0.207
Prior work	Physical	ref.			
	Combination of physical and mental	0.56	0.013	1.107	0.046 *
	Mental	0.468	−0.061	0.997	0.085
	No	0.221	−0.389	0.831	0.479
Number of chronic diseases		−0.421	−0.558	−0.285	<0.001 *
BMI	[kg/m^2^]	0.009	−0.016	0.035	0.475
Physical activity	No	ref.			
	Rare	0.267	−0.074	0.609	0.127
	Yes	0.668	0.353	0.983	<0.001 *
**Walking Speed at a 4 m Distance**					
**Feature**		**Parameter**	**95%CI**		** *p* **
Gender	Women	ref.			
	Men	0.083	−0.197	0.362	0.563
Age	[years]	−0.015	−0.028	−0.002	0.03 *
Education	No, primary	ref.			
	Vocational	0.078	−0.285	0.441	0.674
	Secondary	−0.241	−0.649	0.167	0.249
	High	−0.035	−0.777	0.707	0.926
Prior work	Physical	ref.			
	Combination of physical and mental	0.339	−0.179	0.858	0.201
	Mental	0.202	−0.3	0.704	0.431
	No	−0.035	−0.613	0.544	0.907
Number of chronic diseases		−0.38	−0.51	−0.251	<0.001 *
BMI	[kg/m^2^]	−0.002	−0.027	0.023	0.874
Physical activity	No	ref.			
	Rare	0.313	−0.011	0.637	0.06
	Yes	0.904	0.605	1.202	<0.001 *

Legend: *p*-multifactorial linear regression, * statistically significant difference (*p* < 0.05).

## Data Availability

The data presented in this study are available on request from the corresponding author.

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
