# Peer review of "Assessment of Physical Fitness and Risk Factors for the Occurrence of the Frailty Syndrome among Social Welfare Homes’ Residents over 60 Years of Age in Poland"

_ijerph, 2022, doi:10.3390/ijerph19127449_

Round 1

Reviewer 1 Report

English language and style should be improved, aspecialy in introduction.
Attention must be paid to spelling check. There is few polish words in the tekst.
The descriptions should be shorted by not duplicating the same data.

Reviewer 2 Report

General comments:

The introduction of the manuscript is generally well-written. The authors reported relevant and up to date literature. However, some important information are missing. My main concerns are listed below.

Specific comments

-        It is not clear how the authors defined their thresholds to define participants as frail or pre-frail based on the SPPB. One of the cited literature stated “For pre-frail state versus non-frail, the discriminatory capacity of SPPB was very low, all AUC less than 0.6 (data not shown).”1 It is problematic to use these thresholds without clear description of their validity.

1Using the Short Physical Performance Battery to screen for frailty in young-old adults with distinct socioeconomic conditions

-        Please control your flow-diagram. It seems that several participants are missing in the second box.

-        The authors have to discuss the low power of their study. They stated that 382 participants are needed as a minimum study sample (line 131). However, only 198 participants were analyzed.

-        Even if correlations are statistically significant, correlation coefficients of <0.3 show only weak associations. Please discuss these findings carefully.

Reviewer 3 Report

In the study, the authors showed the high prevalence of frailty among the elderly living in DPS. They also presented that physical activity level as well as age and the number of chronic diseases was an independent predictor for frailty assessed by SPPB score. The paper includes some implications, there still are major concerns.

The introduction section is too long. Same contents are just repeated. It should be more concise.

The authors mentioned that there were few trials assessing frailty using the SPPB battery among DPS residents and that it was the strong point of the study. However, there actually are number of previous research exploring predictors of frailty actually and many of such research include interventional trials.

As the authors mentioned, the cross-sectional design of the study could not prove the causal association between frailty and place of residence or physical activity level. The results of the study do not seem to bring about new, attractive findings.

Most of the discussion section was not based on their own results.

It is not clear what the authors wanted to suggest when they achieved the conclusion that the DPS residents had low physical function and the high prevalence of frailty.

The authors showed that only balance was correlated to the character of prior work. What does the result suggest? There are many previous reports available which found the effect of dual-task training on the fall risk reduction or on cognitive function of elderly people (Lundin-Olsson L, et al. Lancet 349:617, 1997; Yokoyama H, et al. BMC Geriatrics 15:60, 2015). At least, the authors should describe the examples of “mental work”, “combination work”, etc.

Round 2

Reviewer 2 Report

The authors improved the manuscript accordingly.

Reviewer 3 Report

In the revised version, the introduction section appears to become concise. 

This time the authors could discuss based on their own results.

I think the authors well addressed the comments.